# A Maize CBM Domain Containing the Protein ZmCBM48-1 Positively Regulates Starch Synthesis in the Rice Endosperm

**DOI:** 10.3390/ijms23126598

**Published:** 2022-06-13

**Authors:** Xiaojian Peng, Wei Yu, Yirong Chen, Yingli Jiang, Yaru Ji, Long Chen, Beijiu Cheng, Jiandong Wu

**Affiliations:** Anhui Province Key Laboratory of Crop Biology, College of Life Science, Anhui Agricultural University, Hefei 230036, China; 18899590097@163.com (W.Y.); chenyirong0515@163.com (Y.C.); jiangyingli@ahau.edu.cn (Y.J.); jiyaru0608@163.com (Y.J.); cl16720019@163.com (L.C.); cbj@ahau.edu.cn (B.C.)

**Keywords:** maize, *ZmCBM48-1*, endosperm, starch synthesis, expression profiles

## Abstract

Starch directly determines the grain yield and quality. The key enzymes participating in the process of starch synthesis have been cloned and characterized. Nevertheless, the regulatory mechanisms of starch synthesis remain unclear. In this study, we identified a novel starch regulatory gene, *ZmCBM48-1*, which contained a carbohydrate-binding module 48 (CBM48) domain. *ZmCBM48-1* was highly expressed in the maize endosperm and was localized in the plastids. Compared with the wild type lines, the overexpression of *ZmCBM48-1* in rice altered the grain size and 1000-grain weight, increased the starch content, and decreased the soluble sugar content. Additionally, the transgenic rice seeds exhibited an alterant endosperm cell shape and starch structure. Meanwhile, the physicochemical characteristics (gelatinization properties) of starch were influenced in the transgenic lines of the endosperm compared with the wild type seeds. Furthermore, *ZmCBM48-1* played a positive regulatory role in the starch synthesis pathway by up-regulating several starch synthesis-related genes. Collectively, the results presented here suggest that *ZmCBM48-1* acts as a key regulatory factor in starch synthesis, and could be helpful for devising strategies for modulating starch production for a high yield and good quality in maize endosperm.

## 1. Introduction

Maize is one of the most vital staple crops worldwide, and can provide a major energy resource consumption for humans and livestock. Starch accounts for around 75% of the dry matter in the maize endosperm and directly determines the maize quality and yield [1]. With the genome sequence having been completed, maize is an important model species for genetics research. Therefore, more and more research has focused on the starch metabolism and its regulatory mechanism in the maize endosperm.

Starch is synthesized in the plastids of photosynthetic and endosperm cells [2]. In endosperm cells, starch universally consist of two components, linear amylose and branched amylopectin based on the structure. Amylose accounts for around 25% of the total starch, and is a linear polymer with a α-(1,4)-linked D-glucose unit, while amylopectin comprises about 75% of the total starch content and is a highly branched molecule that consists of approximately 95% α-(1,4)-bonds and 5% α-(1,6)-linkages. The ration of amylose/amylopectin directly influences the usage and quality of starch [3]. Starch synthesis is a complex biochemical regulation process, controlled by a series of enzymes, including ADP glucose pyrophosphorylase (AGPase), starch synthase (SS), starch branching enzyme (SBE), and starch debranching enzyme (DBE) [4,5]. ADPase is the rate-limiting enzyme in the process of starch synthesis that is composed of two subunits encoded by the Shrunken 2 (Sh2) and Brittle 2 (Bt2) genes [6]; it is responsible for the first step of starch synthesis. Additionally, GBSS is primarily responsible for amylose chain elongation, while SS, SBE, and DBE catalyze the amylopectin synthesis and modification in the rice endosperm [7]. Moreover, increasing evidence has shown that these starch synthetic enzymes work together to regulate the starch metabolism by forming a stable molecular structure [8,9,10,11]. In the maize endosperm, ISA2 and SSIII might form stable physical complexes in the process of regulating starch synthesis.

Plenty of research has indicated that transcription factors (TFs) play an important role in starch synthesis. According to a report, *OsZIP58* regulates the starch synthesis in rice by changing the amylose content [12]. *ZmDOF36*, a member of the DOF gene family, positively regulates the starch synthesis in maize [13]. The overexpression of *ZmbZIP22* was proven to alter the starch content and structure of maize and rice [14]. *ZmZHOUPI*, which belongs to a bHLHTF, plays a crucial role in kernel development [15]. *ZmbZIP9* binds the promoter cis-elements of starch synthesis-related genes to regulate starch synthesis in maize [16].

In addition to the transcription factors (TFs), it has also been reported that carbohydrate-binding modules (CBMs) are involved in starch metabolism. CBMs are a type of non-catalytic structural domain that mediate the binding of proteins to polysaccharides, such as cellulose, chitin, and starch [17]. The carbohydrate binding module 48 (CBM48) family is one of the largest families of CBMs, which has been identified as a specialized starch-binding domain in recent reports [18,19]. For instance, AtPTST1 and AtPTST2, which also possess a CBM48 domain, have been proven to have key roles in leaf starch metabolism [20,21]. *FLO6* encodes a CBM48 domain-containing protein that is involved in starch granule formation and synthesis in rice [22]. Similarly, another rice CBM48 domain-containing protein, GBSS -BINDING PROTEIN (OsGBP), has been demonstrated to participate in starch biosynthesis by interacting with GBSS1 [23]. However, the CMB48 domain-containing protein in maize is less reported, and whether the protein possesses the CBM48 domain involved in starch biosynthesis is still unclear.

Here, a CBM48-containing gene, which was named *ZmCBM48-1* (*GRMZM2G122274*), from the maize genome was isolated in this study. Our results indicate that *ZmCBM48-1* was specifically expressed in the maize endosperm and was localized in the plastids. The overexpression of *ZmCBM48-1* in rice can change the grain thickness and 1000-grain weight. Importantly, the starch content, starch structure, and physicochemical characteristics of starch were influenced in the *ZmCBM48-1*-overexpressing rice endosperm. Compared with the wild type, the expression patterns of the starch synthesis-related genes were also regulated in the transgenic rice endosperm. Collectively, these results suggested that *ZmCBM48-1* may perform essential functions in the process of maize starch metabolism, and could help to devise strategies for modulating starch production for a high yield and superior quality in the maize endosperm.

## 2. Results

### 2.1. Isolation and Sequence Analysis of ZmCBM48-1

The carbohydrate binding module 48 (CBM48) family is one of the largest families of CBMs, which has been identified as a specialized starch-binding domain in recent reports [18,19]. However, the CBM48 member and its function in maize is still unclear. We searched the maize genome using the CBM48 conserved domain using the BLASTp method, and identified a CBM48-containing protein, which was named ZmCBM48-1 (GRMZM2G122274). ZmCBM48-1 contained a typical CBM48 domain and a coiled-coil (CC) domain in the C-terminal region (Figure 1a,b). The phylogenetic tree was constructed to explore the relationship between ZmCBM48-1 and the homologs in other higher plants. The results showed that ZmCBM48-1 had a high similarity to other plant CBM48 members, such as FLO6 (a rice CBM48 domain protein) (Figure 1c), which suggests that ZmCBM48-1 might be involved in starch metabolism.

### 2.2. Expression Patterns Analysis of ZmCBM48-1 in Different Maize Tissues

The expression patterns of *ZmCBM48-1* from different maize tissues were detected through publicly available genome-wide transcript profiling. *ZmCBM48-1* had a higher expression in the maize endosperm than the other tissues (Figure 2a). Moreover, different maize tissue samples, including root, stem, leaf, ear, silk, tassel, and embryo, and 6 DAP endosperm, 9 DAP endosperm, 12 DAP endosperm, 15 DAP endosperm, 18 DAP endosperm, 21 DAP endosperm, and 24 DAP endosperm were collected to analyze the temporal and spatial expression profiles. The results showed that *ZmCBM48-1* had a high expression in the maize endosperm (Figure 2b,c), suggesting that *ZmCBM48-1* may be regulated in endosperm development in maize.

### 2.3. ZmCBM48-1 Localizes in Plastids

To examine the sub-cellular localization of ZmCBM48-1, the 35S::ZmCBM48-1::GFP was constructed (Figure 3a). The transient expression of ZmCBM48-1 that was constructed was transiently expressed in the maize leaves’ protoplasts and was observed under a confocal microscope. The results showed that the GFP signal of the ZmCBM48-1-GFP fusion protein co-localized with the chlorophyl II autofluorescence signal in the chloroplasts (Figure 3b), which revealed that ZmCBM48-1 was localized in the plastid.

### 2.4. Ectopic Expression of ZmCBM48-1 Alters Grain Morphology

First, 16 independent *ZmCBM48-1*-overexpressing rice lines were obtained via the *Agrobacterium* mediated transformation method, and two homozygous T3 transgenic lines (L4 and L7) possessing different mRNA expression levels were used to investigated the function of ZmCBM48-1 in rice seed development (Figure 4a,b). The grain length, width, thickness, and 1000-grain weight were measured in both wild type (WT) and *ZmCBM48-1*-overexpressing rice lines. Compared with the WT lines, the overexpression of ZmCBM48-1 did not show any visible difference between grain length and grain width in the transgenic rice lines (Figure 4c,d). However, the grain thickness and 1000-grain weight were decreased in the *ZmCBM48-1*-overexpressing transgenic rice lines (Figure 4e,f). These results indicate that ZmCBM48-1 could influence grain development in rice.

### 2.5. Overexpression of ZmCBM48-1 Influences the Starch Content and Starch Structure in Rice

To further investigate whether *ZmCBM48-1* has a direct impact on starch synthesis, we tested the amylose content, amylopectin content, and total starch content in the WT and transgenic lines. The results showed that the total starch content, amylose content, and amylopectin content were higher in the transgenic rice endosperm than those of the WT lines (Figure 5a–c). The soluble sugar content was measured, which has been reported to participate in starch accumulation [24]. In contrast, the soluble sugar content in the transgenic lines was dramatically decreased (Figure 5d). Moreover, the protein content of the endosperm was higher in the transgenic lines than that of the WT lines (Figure 5e). These results indicate that *ZmCBM48-1* influenced the starch content and composition in the rice endosperm. This conclusion was also further demonstrated by the results of the BV (blue value) and the maximum absorption wavelength (λmax). The BV and λmax of the amylose and amylopectin were indeed increased in the transgenic lines compared with the WT lines (Figure 6a,b).

BV and maximum λmax can provide indicators for the starch structure and represent the ability of starch binding to iodine [25]; so, we carried out the histological analysis and scanning electron microscopy (SEM) analysis to observe the endosperm development and starch structure. The results showed that the overexpression of rice lines exhibited larger and looser endosperm cells that were composed of loosely packed starch granules (Figure 6c and Figure 7). In contrast, WT seeds possessed small endosperm cells and were filled with densely packed starch granules (Figure 6c). Collectively, these results reveal that *ZmCBM48-1* changed the starch structure in the rice endosperm.

### 2.6. Enhanced the Expression of Starch Synthesis-Related Genes in ZmCBM48-1 Transgenic Plant

To examine the regulatory role of *ZmCBM48-1* in the expression of previously identified starch synthesis genes, the expression profiles at different development stages for 16 starch biosynthesis-related genes were analyzed in the rice endosperm of transgenic and WT lines using quantitative RT–PCR. The expression level of the detected genes, including *OsAGPL1, OsAGPL2, OsAGPS1, OsAGPS2a, OsSSI, OsSSIIa, OsGBSSI, OsBEIIa,* and *OsISA1,* were consistently upregulated through the entire stage of the *ZmCBM48-1* transgenic lines (Figure 8). These results reveal that the *ZmCBM48-1* gene played an extensive role in the regulation of the starch metabolism in the endosperm.

## 3. Discussion

CBM48-containning proteins have been reported to bind the starch in plants and animals [26,27]. In *Arabidopsis*, AtPTST1 and AtPTST2, which also contain a putative CBM48 domain, have been proven to play key roles in leaf starch metabolism [20,21]. Similarly, two CBM48 domain-containing proteins, FLO6 and GBSS -BINDING PROTEIN (OsGBP) in rice, which are also localized in plastids, have been demonstrated to be involved in regulating starch biosynthesis by interacting with ISA1 and GBSS1, respectively [22,23]. In mammals, a catalytic α-subunit and accessory β- and γ-subunits form a heterotrimeric complex, namely AMPK [28]. Among them, the β subunit contains a conserved glycogen-binding domain (GBD, also named CBMs in plants), which can bind glycogen [18]. In this study, we isolated a CBM48-containing protein from maize, ZmCBM48-1, which contained the conserved amino acid residues, based on the results of a multiple sequence alignment and phylogenetic analysis. In addition, ZmCBM48-1 was specifically expressed in the maize endosperm and was localized in the plastids, which us consistent with its features in endosperm starch synthesis.

Starch consists of amylose and amylopectin and has important application value in the economy and industry. Starch synthesis is a complex metabolic process that requires the coordinated activities of a series of enzymes [4]. To investigate the function of *ZmCBM48-1* in starch metabolism, transgenic rice lines overexpressing *ZmCBM48-1* were obtained. The soluble sugar content in the transgenic rice seed endosperm was lower than that of the WT seeds, which suggests that it was more soluble sugar have been converted into starch in the transgenic lines. This result can partially explain the data that the overexpression of *ZmCBM48-1* in rice increased the starch content compared with the WT lines. There is approximately 10% protein and 70% starch in the maize endosperm [29]. Interestingly, the protein content also increased in the transgenic rice endosperm. These results suggest that *ZmCBM48-1* can affect endosperm development. This was also supported by our histological analysis results, that the overexpression of rice lines exhibited larger and looser endosperm cells. In addition, the amylose and amylopectin content were also increased in the transgenic line seeds, which was also further supported by the data of the blue value (BV) and λmax. Previous studies have shown that the BV and λmax values can provide a key indicator of the starch content and structure, and represent the ability of starch binding to iodine [25]. Moreover, the increased BVs of amylose at λmax 620 nm and BVs of amylopectin at λmax 590 nm in transgenic lines also indicated that the starch structure could also be altered. As expected, the starch structure was indeed changed in the transgenic rice endosperm. The SEM results showed that the overexpression of *ZmCBM48-1* exhibited loosely packed starch granules compared with the WT lines. These results demonstrate that *ZmCBM48-1* indeed regulates starch metabolism in the seed endosperm.

Starch synthesis is catalyzed by four types of enzymes: AGPase, SS, SBE, and DBE [4,5]. Previous research has shown that FLO6 and OsGBP could regulate starch synthesis by interacting with starch-related enzymes ISA1 and GBSS1, respectively [22,23]. To further analyze the regulatory function of *ZmCBM48-1* in starch synthesis, the expression profiles of nine starch synthesis-related genes were characterized in the transgenic line endosperm. The results indicated that the expression patterns of these genes were upregulated in the *ZmCBM48-1*-overexpressing transgenic lines. Obviously, further research, including screening the direct interaction proteins with ZmCBM48-1 and the production of transgenic maize lines, will be needed in future research.

In summary, our study demonstrated the function of *ZmCBM48-1* in rice. The overexpression of *ZmCBM48-1* in rice increased the starch content and changed the starch composition and starch structure compared with the WT lines. In addition, the gelatinization properties of the endosperm starch also influenced the transgenic lines. Moreover, the expression profiles of the starch synthesis genes were also upregulated in the transgenic lines. These results suggest that ZmCBM48-1 is a potential regulatory factor in starch synthesis, but whether it maintains the same function in maize is still unknown. In future research, we will screen the interaction protein and produce the overexpression and knockout methods in the maize line to clarify the regulatory mechanism of *ZmCBM48-1*. More importantly, ZmCBM48-1 regulated downstream genes will be screened to explore whether they participate in the metabolic regulation network. Future work will discover the molecular mechanism of ZmCBM48-1 and provide further insight into high-yield and high-quality maize breeding.

## 4. Materials and Methods

### 4.1. Plant Materials and Growth Conditions

The maize (B73) seedlings were grown at 26–28 °C for a day length of about 13.5–14 h and with 65% air humidity in a greenhouse. Wild type (WT) rice (Zhonghua11) and transgenic rice lines were grown under natural conditions.

### 4.2. Multiple Sequence Alignment and Phylogenetic Analysis

The sequence analysis was performed as described previously [13]. Briefly, the amino acid sequences of the CBM48-containing domain proteins from different plants were investigated using ClustalX software, and the CBM48 conserved motifs were defined using Pfam and SMART. The phylogenetic tree was constructed using MEGA (version 4.0, Auckland, New Zealand) based on the neighbor joining (NJ) method.

### 4.3. Total RNA Extraction and qRT-PCR

Maize samples of the root, stem, leaf, ear, silk, tassel, and embryo, and 6 DAP (days after pollination) endosperm, 9 DAP endosperm, 12 DAP endosperm, 15 DAP endosperm, 18 DAP endosperm, 21 DAP endosperm, and 24 DAP endosperm were collected for RNA extraction, separately. RNA extraction was performed according to the manufacturer’s protocol of the RNAiso plus kit (TaKaRa, Dalian, China), then SuperScript^TM^ III reverse transcriptase (Invitrogen) was used for generating the cDNA. The qRT-PCR was carried out using procedures described previously [30]. The value of maize *Actin1* was used as a control for the expression analysis. To analyze the expression of the starch synthesis-related genes, samples of 6 DAP endosperm, 9 DAP endosperm, 12 DAP endosperm, and 18 DAP endosperm of WT and transgenic rice lines (mixed samples of different transgenic lines) were also collected for the qRT-PCR test. The rice *OsTublin* gene was used as a reference gene (Table 1). All of the experiments were quantified in three replicates.

### 4.4. Production of ZmCBM48-1-Overexpressing Transgenic Rice Lines

The open reading frame (ORF) of *ZmCBM48-1* (*GRMZM2G122274*) was inserted into the pCAMBIA1301 vector, then the construct was introduced into *O. sativa japonica cv.* Zhonghua11, as described previously [13]. We obtained 16 independent *ZmCBM48-1*-overexpressing lines, and two homozygous T3 transgenic lines (L4 and L7) were identified for further analysis.

### 4.5. Subcellular Localization of ZmCBM48-1

The full length of *ZmCBM48-1* was amplified with specific primers (listed in Table 1) and was inserted into pCAMBIA1305 containing the GFP driven by the CaMV 35S promoter. Then, the clone was transformed into maize leaf protoplasts separately according to the previous report [31]. After two-day transformation, a confocal laser scanning microscope (Zeiss LSM 780) was used to observe the fluorescence.

### 4.6. Determination of Agronomic Characters of Transgenic Rice

Rice length, width, and 1000-grain weight from 1000 randomly chosen seeds from fully filled grains of WT and transgenic lines were measured using an A3 scanner (UniscanM1, Shanghai, China). The fully filled grain thicknesses were detected using a vernier caliper from 200 randomly chosen seeds.

### 4.7. Measurement of Starch Properties

The total starch contents of the rice endosperm were measured using the starch assay kits following the instructions provided (K-TSTA, Megazyme). The iodine colorimetric method (K-AMYL, Megazym) was used to detect the amylose content. To determine the amount of amylopectin, the amylopectin assay kit was performed following the manufacturer’s protocol (A152-2-1, Jcbio, Nanjing, China). The soluble sugar was measured following the method by [12]. The total protein was detected using a previously reported method [32]. The alkali impregnation method was used in the analysis for detecting the blue value (BV) and maximum absorption wavelength (λmax) [33].

### 4.8. Histological Analysis of Transgenic Rice

First, 18 DAP rice seeds of transgenic and WT lines were collected for the analysis of seed development, following [34]. Briefly, the seeds were fixed in FAA (formalin–aceto–alcohol) and were dehydrated in a series of graded ethanol, and then embedded into paraffin and polymerized for 3 days at 37 °C. After being sectioned into 4 μm thick sections, the samples were stained with toluidine blue and were observed with a light microscope (Nikon, Japan).

### 4.9. Scanning Electron Mmicroscopy (SEM)

To detect the starch structure of the rice seeds, SEM (Hitachi, Tokyo, Japan) was used based on research from a previous study [13].

### 4.10. Data Analysis

Values are shown as mean ± standard deviation (SD). Statistical analyses were performed using SPSS v16.0 software based on Student’s *t*-test.

## 5. Conclusions

In this study, we identified a novel starch regulatory gene, ZmCBM48-1, which contains a carbohydrate-binding module 48 (CBM48) domain. ZmCBM48-1 was highly expressed in maize endosperm and localized to the plastids. Compared with the wild type lines, overexpression of ZmCBM48-1 in rice altered the grain size and 1000-grain weight, increased the starch content and decreased the soluble sugar content. *ZmCBM48-1* positively regulate the starch synthesis by upregulating the expression levels of starch synthesis related genes.

## Figures and Tables

**Figure 1 ijms-23-06598-f001:**
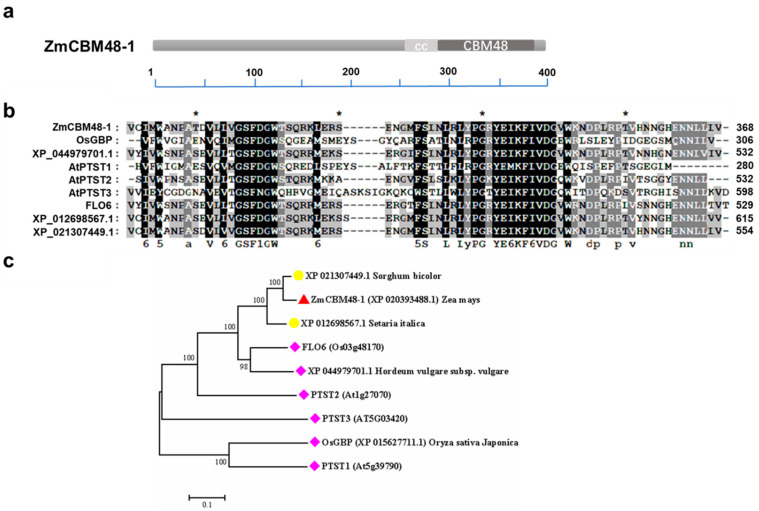
ZmCBM48-1 is a CBM48 domain-containing protein. (**a**) ZmCBM48-1 contains a CBM48 domain and coiled-coil (CC) domain. (**b**) Multiple sequence alignment of CBM48-containing sequences. The conserved residues of the CBM48 domain are marked with a star. (**c**) A phylogenetic tree was constructed using MEGA4.0 based on the N-J method. Bootstrap values from 1000 replicates are indicated at each node.

**Figure 2 ijms-23-06598-f002:**
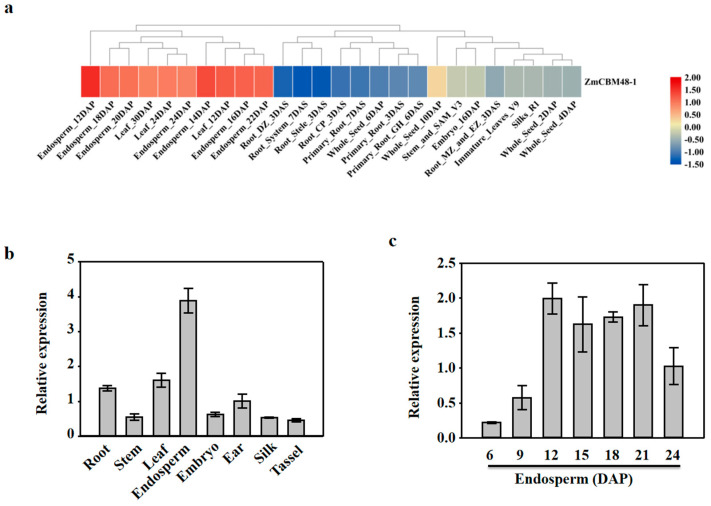
Expression profile analysis of the *ZmCBM48-1* gene. (**a**) Expression profiles of *ZmCBM48-1* in different tissues. Red, yellow, and white indicate high, medium, and low levels of gene expression, respectively. (**b**,**c**) Tissue-specific expression pattern analysis of *ZmCBM48-1* in various maize tissues. The *ZmActin1* gene was used as an internal control. Error bars are the standard deviations of three technical repeats and two biological repeats. DAP, days after pollination.

**Figure 3 ijms-23-06598-f003:**
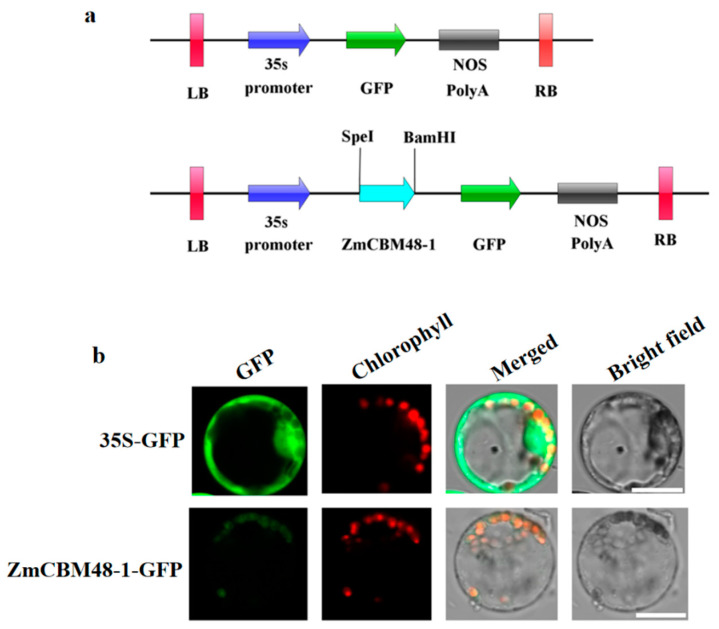
Subcellular localization of ZmCBM48-1. (**a**) Schematic representation of the ZmCBM48-1-GFP fusion construct and the empty vector (GFP) used for the transient expression. (**b**) The fusion proteins and 35S::GFP DNA constructs were transiently expressed separately in themaize leaf protoplasts and were visualized under a confocal laser scanning microscope. Scale bar, 10 µm.

**Figure 4 ijms-23-06598-f004:**
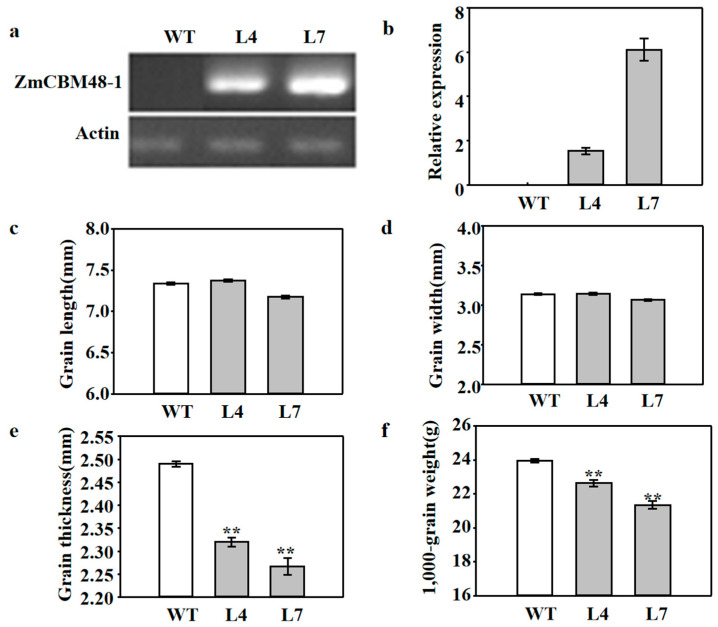
Agronomic character analysis of the *ZmCBM48-1* transgenic rice lines. (**a**,**b**) Molecular identification of *ZmCBM48-1* transgenic rice plants. WT: wide type, L4 and L7 represent the different transgenic lines, *OsA**ctin1* was used as an internal control. (**c**) Length of seeds. (**d**) Width of seeds. (**e**) Thickness of seeds. (**f**) The 1000-grain weight of the seeds. Data are presented as means ± SD from three replicates. ** represents significant differences between WT and transgenic plants at *p* < 0.01.

**Figure 5 ijms-23-06598-f005:**
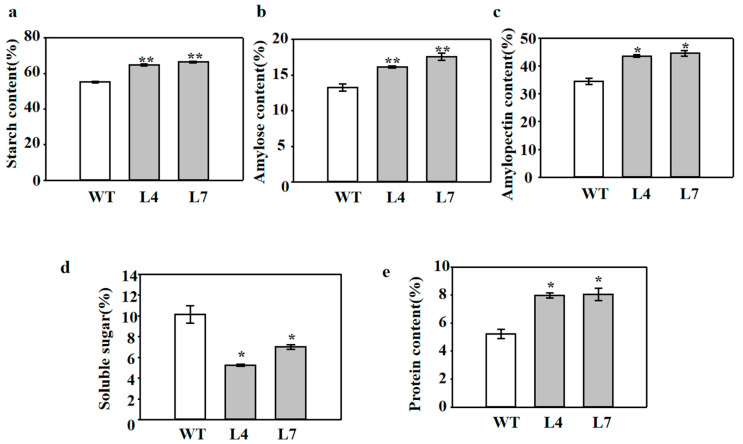
The overexpression of *ZmCBM48-1* in rice altered the starch content. (**a**) Total starch content. (**b**) Amylose content. (**c**) Amylopectin content. (**d**) Soluble sugar content. (**e**) Protein content. Data are presented as means ± SD from three replicates. ** and * represent significant differences between WT and transgenic plants at *p* < 0.01 and *p* < 0.05, respectively.

**Figure 6 ijms-23-06598-f006:**
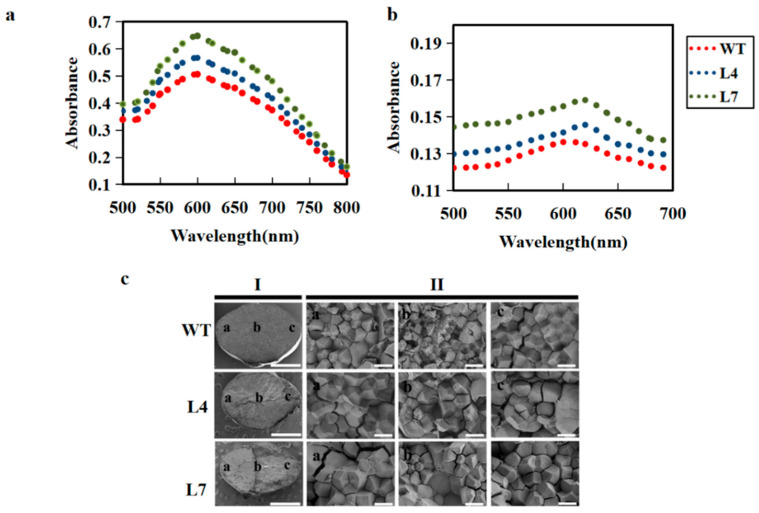
The overexpression of *ZmCBM48-1* in rice influenced the starch structure and gelatinization properties. (**a**) Absorbance spectra of the isolated amylose-iodine complexes. BV at 620 nm and λmax represent the ability of amylose to combine with iodine. (**b**) Absorbance spectra of the isolated amylopectin–iodine complexes. BV at 590 nm and λmax represent the ability of amylopectin combined with iodine. (**c**) Scanning electron microscopy (SEM) analysis of the starch structure from mature kernels of the wild type and *ZmCBM48-1* transgenic seeds. The ventral area, central area, and dorsal area of the mature endosperm are indicated in a, b, and c (**c**). Scale bars: 1 mm in (I); 10 μm in (II).

**Figure 7 ijms-23-06598-f007:**
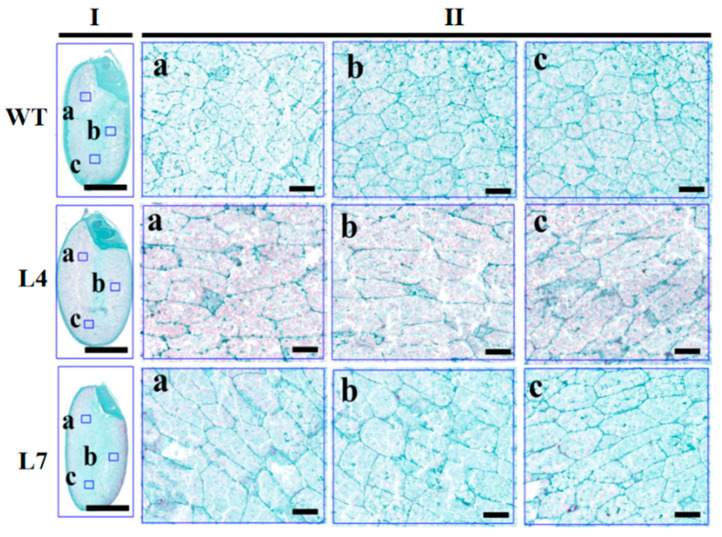
Paraffin sections of the endosperm of WT and transgenic rice seeds. (**a**–**c**) The endosperm cells in different sections. Scare bars: 2 mm in I, 50 μm in II.

**Figure 8 ijms-23-06598-f008:**
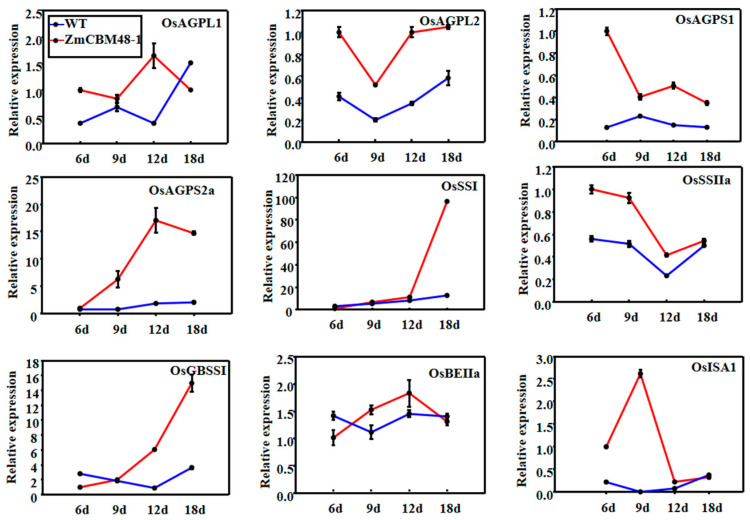
Expression patterns of 16 starch synthesis-related genes at 6, 9, 12, and 18 days after pollination (DAP). The mRNA expression levels of each gene in the six DAP seeds of the wild type (WT) were used as a control.

**Table 1 ijms-23-06598-t001:** Primers used in this work.

Assay	Primer Name	Sequence (5′-3′)
qRT-PCR analysis	ZmActin-F	CTGACGGAGCGTGGTTACTCAT
	ZmActin-R	TGGTCTTGGCAGTCTCCATTTC
	ZmCBM48-1-F	AGTTTATCGTTGACGGTGTTTG
	ZmCBM48-1-R	AACTTCAAGTCAAGTGACAAGC
	OsTublin-F	TACCGTGCCCTTACTGTTCC
	OsTublin-F	CGGTGGAATGTCACAGACAC
	OsAGPL1-F	GGAAGACGGATGATCGAGAAAG
	OsAGPL1-R	CACATGAGATGCACCAACGA
	OsAGPL2-F	AGTTCGATTCAAGACGGATAGC
	OsAGPL2-R	CGACTTCCACAGGCAGCTTATT
	OsAGPS1-F	GTGCCACTTAAAGGCACCATT
	OsAGPS1-R	CCCACATTTCAGACACGGTTT
	OsAGPS2a-F	AACAATCGAAGCGCGAGAAA
	OsAGPS2a-R	GCCTGTAGTTGGCACCCAGA
	OsBEIIa-F	GCCAATGCCAGGAAGATGA
	OsBEIIa-R	GCGCAACATAGGATGGGTTT
	OsGBSSI-F	AACGTGGCTGCTCCTTGAA
	OsGBSSI-R	TTGGCAATAAGCCACACACA
	OsSSI-F	GGGCCTTCATGGATCAACC
	OsSSI-R	CCGCTTCAAGCATCCTCATC
	*OsSSIIa-F*	GCTTCCGGTTTGTGTGTTCA
	*OsSSIIa-R*	CTTAATACTCCCTCAACTCCACCAT
	*OsIAS1-F*	TGCTCAGCTACTCCTCCATCATC
	*OsIAS1-R*	AGGACCGCACAACTTCAACATA
Subcellular localization		
	ZmCBM48-1-GFP-F	GG ACTAGT ATGCCACCGCGTCCCGCGCT
	ZmCBM48-1-GFP-R	CGGGATCCAGTGACAAGCAGAAGGTTGTTTTCA

## Data Availability

All data are displayed in the manuscript.

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
