# Peer review of "A Maize CBM Domain Containing the Protein ZmCBM48-1 Positively Regulates Starch Synthesis in the Rice Endosperm"

_ijms, 2022, doi:10.3390/ijms23126598_

Round 1

Reviewer 1 Report

220508_Int. J. Mol. Sci. 2022 review

Carbohydrate-binding modules are non-catalytic structural domains that mediate the binding of proteins to polysaccharides, such as cellulose, chitin, and starch. Out of them, the CBM48 family is one of the most prominent families of CBMs, which has been identified as a specialized starch-binding domain. The characterization of CBM48-containing proteins has been conducted in Arabidopsis, rice, barley, and cassava. However, the CMB48 domain-containing protein in maize was less reported and characterized. Therefore, the authors focus on ZmCBM48-1, one of the CBM48-containing proteins in maize in this paper.

Their analysis is very straightforward, cloning ZmCBM48-1 cDNA, overexpressing it in rice, and confirming its phenotype. It was not mentioned why rice was used instead of maize, but it is probably because it is easier to handle. However, it is necessary to mention the possibility that the function of ZmCBM48-1 in rice is different from in maize.

With the following improvements, this paper could be appropriate to be published in International Journal of Molecular Science.

==Major points==

1.

The phylogenetic tree in Figure1b shows that ZmCBM48-1 are orthologous genes of rice FLO6 and Arabidopsis AtPTST2. However, the protein length of ZmCBM48-1 (367aa) is shorter than those of FLO6 (530aa) and AtPTST2 (532aa). It is possible that the isolated cDNA of ZmCBM48-1 is NOT full length and may lack the N' terminal coding region. Please conduct the 5' RACE experiment to determine the 5' end of ZmCBM48-1 cDNA.

If the isolated one were a full-length cDNA, the authors should discuss why ZmCBM48-1 is shorter than FLO6 and AtPTST2.

2.

Amino acid number (598) of AtPTST2 in Figure1b may be wrong. AtPTST2 is a 532 amino-acid protein. Please check the amino acid numbers of other proteins, too.

3.

In Figure 3b, the GFP signal of ZmCBM48-1-GFP fusion is very faint. Is it possible that non-transformed cells would have the same level of signals as the background at the same exposure time? Negative control (non-transformed cells) and positive control (another chloroplast-targeted GFP) should be taken under the same imaging conditions.

4.

L219

What is the BV (blue value)?

What kind of starch properties are reflected by BV?

Could you add the comments to explain this?

5.

CBM48-containing protein has been reported to interact with starch-related enzymes, and overexpression of ZmCBM48-1 may disrupt these endogenous interactions. Therefore, the original function of ZmCBM48-1 is challenging to predict from the phenotypes of the rice transformant in this study. Therefore, the authors should mention the difficulty in interpreting the results of this study.

-----------------------

==Minor points==

L27

75% =>  around 75%

L34

25% =>  around 25%

L51

 According to the report (italic), => According to the report (regular),

L51

was (italic) => was (regular)

L260

plans => plants

Sicerely,

Reviewer 2 Report

A maize CBM domain containing protein, ZmCBM48-1, positively regulate the starch synthesis in rice endosperm

In the present manuscript, the author focuses on how ZmCBM48-1 gene influences starch metabolism in rice and starch properties in transgenic plants.

I read this manuscript with interest. But I think it should be improved a little bit.

I have a few questions/comments:

1.

I suggest to rewrite Abstract (very little changes):

Starch directly determines the grain yield and quality. The key enzymes participating in the process of starch synthesis have been cloned and characterized. Nevertheless, the regulatory mechanisms of starch synthesis remain unclear. In this study, we identified a novel starch regulatory gene, ZmCBM48-1, which contains a carbohydrate binding module 48 (CBM48) domain. ZmCBM48-1 was highly expressed in maize endosperm and was localized in the plastids. Compared with the wild-type lines, overexpression of ZmCBM48-1 in rice altered the grain size and 1000-grain weight, increased the starch content, and decreased the soluble sugars content. Additionally, the transgenic rice grains exhibited altered endosperm cell shape and starch structure. At the same time, the physicochemical properties (gelatinization properties) of starch in the endosperm of the transgenic lines were affected compared to the wild-type seeds. Moreover, ZmCBM48-1 plays a positive role in regulating the starch synthesis pathway by up-regulating several starch synthesis-related genes. Overall, the results presented here suggest that ZmCBM48-1 is an important regulatory factor in starch synthesis and may be useful in developing strategies to modulate starch production for high yield and good quality in maize endosperm.

2.

Line129: Only one soluble sugar was measured with method described in Wang, et al 2013?

Is it possible with this method to determine the amounts of sucrose, fructose, glucose?

How the Authors understand “soluble sugar”, which carbohydrates could it be?

3.

there are some spelling mistakes in the text (ex: line 224) “sugar contetn” -  please correct all

  1.  

Figure 4 (b-f) should be redone, the scales on the Y axis start not with “0” - why? If the differences are statistically significant, there is no need to modify the plot.

Figure 5 - Similar remakes:  the scales on the Y axis max value should be the same (80%?) - for Fig 5 a-c. I did not understand: the calculated percentage is converted to dry weight or fresh weight of grains?

5.

I have no doubt that in transgenic plants (lines) the starch structure, the ratio of amylose to amylopectin, the starch content, the protein content, and the content of soluble sugars are altered.

But how do the authors explain the phenomenon that 1000-grain weight and grain thickness decrease in transgenic plants, the amount of starch, amylose, amylopectin, and protein increases, and the endosperm is larger and looser than in wild-type plants - how can this be explained? Maybe lecture e.g. Tetlow and Emes, Agronomy 2017/7/81 will help?

In my opinion, something is missing here ... Does ZmCBM48-1 gene influences only starch metabolism?

For the future - not in this publication - it is worth checking the expression of genes involved in the degradation of starch - not only in its synthesis. The total amount of starch, its property is the result of biosynthesis and degradation of starch granules in the endosperm

Round 2

Reviewer 1 Report

>>Response 2:Thank you for the reminder, we have rechecked the length of these amino acid and found that amino acid number is right. In this study, sequence data of the two genes can be found in TAIR (www.arabidopsis.org) under the following accession numbers: PTST1 (At5g39790), PTST2 (At1g27070). In addition, we also refer to the following literature:

Did you really check? In TAIR website, PTST2 (At1g27070) is as follows,
https://www.arabidopsis.org/servlets/TairObject?type=aa_sequence&id=1009110790

According to this site, PTST2 (At1g27070) is 532 amino acids.
So, your numbering (598) in Figure 1 b apparently to be wrong.

FLO6 has the same error. Please check this one yourself.
Please confirm other proteins, too.

If the errors are not corrected in the next manuscript, I will not accept further peer review.

Round 3

Reviewer 1 Report

Dear Editor and authors,

Congratulations on this excellent work.
Since the authors have appropriately addressed the points I raised, I think it is acceptable for this manuscript to publish in your journal.
However, there seem to be unnecessary borders in figures b and c of Figure 1, which should be removed when editing.

I appreciate for giving me a chance to review this excellent study.
